

# An open source refactoring of the Canadian small lakes model for estimates of evaporation from medium sized reservoirs

M. Graham Clark[1,2], Sean K. Carey[2]

[1]Department of Earth Sciences, St. Francis Xavier University, Antigonish, NS, B2G 2W5, Canada
[2]School of Earth, Environment & Society, McMaster University, Hamilton, ON, L8S 4K1, Canada

*Correspondence to*: M. Graham Clark (dr.mg.clark@gmail.com)

**Abstract.** Eddy covariance (EC) is one of the most effective ways to directly observe evaporation from a lake surface. However, the deployment of EC systems on lakes is costly and technically challenging which engenders a need for accurate modelling of evaporation from reservoirs for effective management. This study aims to 1) refactor the Canadian Small Lakes Model (CSLM) into modern high-level programming languages in open-source repositories and 2) evaluate evaporation estimates from the CSLM using nine years of EC observations of a pit-lake in Northern Alberta. The CSLM is a 1-D physical lake model simulating a mixing layer and arbitrary thick skin layer which interfaces with the atmosphere and includes a module for ice dynamics. It was developed to interface with the Canadian Global Coupled models as part of the surface classification scheme, and thus utilizes widely accessible forcing data. In this study the CSLM evaporation estimates are also compared to a commonly used bulk transfer method of estimating evaporation. In general, the CSLM had smaller open-water season error (RMSE of 0.70 mm day$^{-1}$) than the bulk transfer method (RMSE of 0.83 mm day$^{-1}$). However, if EC data is available, further improvement can be gained by using an Artificial Neural Network to adjust the modelled fluxes (RMSE of 0.51 mm day$^{-1}$). This final step can be very useful for gap-filling missing data from lake observation networks as there has been recent attention on the limited coverage of direct open water evaporation observations in the literature.





## 1 Introduction

Globally, the volume of water stored in reservoirs used for hydroelectricity production as well as water and waste management has been increasing. Managing these reservoirs is complex and one of the largest uncertainties is the quantification of evaporation (Long *et al*., 2007), which is rarely measured. Direct observation of the evaporative fluxes over small and large
bodies of water is uncommon due to logistical difficulty, safety, cost, and power availability. When observations are made, it is typically during the warm open water season with less of an emphasis on the cooling phase of the year when evaporation is greatest (Rouse *et al*.,2005; Blanken *et al*., 2011). The bias in observations of northern lakes towards warm stratified periods results in a limited understanding of the drivers of evaporation variability for reservoir managers. Complete observation records of the ice-free season are important since prior work (e.g. Spence *et al*. 2003; Lenters *et al*. 2005; Granger and Hedstrom,
2011; Spence *et al*. 2011; Liu *et al*. 2012; Shao *et al*. 2020; Fournier *et al*., 2021) has shown significant differences in atmospheric forcing that drive evaporation processes between the warming, cooling, and frozen phases of lakes.  Here warming is broadly defined as the spring mixing and stratification period when the atmosphere is generally warmer than the air, cooling is turnover when the lake is warmer than the air, and frozen is the ice-covered period.

In addition to the management implications, variability in the rate of evaporation has regional impacts on the atmospheric
conditions since bodies of water alter the climatology of landscape (Rouse *et al*. 2005; Long *et al*. 2007; Blanken *et al*. 2011). In landscape scale reclamation projects, such as those in the Athabasca Oil Sands region, variation in surrounding atmospheric conditions directly impacts the landscapes surrounding water bodies and should be considered in the reclamation plan. Lakes contribute to increased humidity and alter the seasonality of surface energy storage and exchange. In general, the mixing dynamics and large heat capacity of water bodies result in a surface that is slower to heat and slower to cool then surrounding
terrain. This differential in the thermal momentum of lakes produces large sensible and latent heat fluxes to the atmosphere during the cooling season.  In large northern lakes, which remain mostly ice-free, up to 88% of the annual evaporative flux occurs over the cold months (Blanken *et al*. 2011).  Even in smaller lakes, or large lakes that completely freeze, the cooling period prior to ice formation represents the most significant period for water loss to the atmosphere (Clark *et al*. 2021). Although studies of direct lake evaporation have been increasing in recent years, the period prior to ice formation remains
difficult to measure due to the complications in both personnel safety and mechanical damage to instruments that lake-ice causes in field-based research programs.

Broadly there are three typical approaches to estimating evaporation from widely available, or easily measured, meteorological data; aerodynamic/bulk transfer methods, radiation-based estimates, and empirical equations (Fournier *et al*. 2021). The bulk transfer method is derived from Dalton's law and Monin-Obukhov similarity theory and is based on the drying power of air as
a function of turbulence and humidity (e.g. Heikinheimo *et al*., 1999). Due to the need to estimate turbulent conditions, this method requires the most localized meteorological observations. Radiation based methods primarily include available energy which is more consistent over large areas especially when surface radiative properties are consistent (e.g. Shuttleworth, 2012) and are accurate at daily flux timescales, or longer (Granger and Hedstrom, 2011).  Empirical equations are typically derived



from regressions to available data (e.g. Granger and Hedstrom, 2011). Both these later methods have more flexible data
requirements than the bulk transfer method, but the trade-off is a greater uncertainty in extrapolating beyond the conditions
for which they were developed. There are several hybrid approaches or "combination equations" but are variations on the
above approaches. In general, due to the distinct differences in drivers of evaporation between the warming and cooling phases,
the most accurate estimates of evaporation come from approaches utilizing localized windspeed and surface temperatures
estimates (Fournier *et al*. 2021). In addition to localized data, this paper aims to also highlight the advantage of using a
processed-based model for estimating evaporation rates from managed northern reservoirs. A mechanistic model can provide
not only evaporation estimates, like the above approaches, but also provides insight into lake processes that could be valuable
to managers and researchers, such as turnover timing(s), ice formation and season length, total energy storage, shear and
buoyant forces, thermal profile, and depth to thermocline.

For both water resource management and for evaluating the impact of climate change there is an emergent need to be able to
accurately quantify the lake-atmosphere flux of water and energy. Here we compare two methods, the Air-Sea Toolbox
developed by Fairall *et al*. (1996) and the Canadian Small Lakes Model (CSLM) developed by Mackay (2012). The first, the
Air-Sea Toolbox, is a bulk flux algorithm developed for Tropical Ocean-Global Atmosphere (TOGA) Coupled Ocean
Atmosphere Response Experiment (COARE). The second, the CSLM, is a one-dimensional dynamic model developed for
inclusion with the Canadian Land Surface Scheme (CLASS) to be used in the CanGCM and Environment and Climate Change
Canada's numerical prediction systems. Here we present 1) the refactoring of the CSLM to high level languages and open-
source distribution for use by scientists and land managers who may have limited modelling expertise, and 2) an assessment
of these two modelling approaches for estimating northern lake-atmosphere exchange with almost a decade of direct eddy
covariance-based lake-surface flux observations.

## 2 Methods

### 2.1 Study site

Base mine lake (BML) is located on the Syncrude Canada Ltd, Mildred Lake mine lease in Northern Alberta Canada (Köppen
climate Dfb, Boreal plains ecozone). BML is a reclaimed pit-lake, built on an active oil sands pit mine. The lake and
surrounding topography was constructed from pit-infill and overburden to be the terminus of a constructed 660 ha watershed
of upland forests and peat filled lowlands. From 1994 to 2012, fluid fine tailings (FFT; by-products of the oil sands mining
process) were stored at the site to a depth of 45 m with an additional 3 – 5 m freshwater capping. In 2012, the water capping
was increased 8 - 11 m to initiate the reclamation of the lake in the freshwater cap. The exact quantity of freshwater varies as
the lake is highly managed with freshwater regularly added and removed during mine operations (Clark *et al*. 2021 for more
details of water management). The FFT have greatly consolidated and "dewatered" into the lake column, producing a semi-
solid bottom layer under the freshwater capping. Alum treatments used to clarify BML in late 2016 led to the formation of a
precipitated fine-sediment layer between the freshwater and FFT, sometimes referred to as the "mud layer". This layer is





regularly re-mixed within the water column leading to large variability in the turbidity of BML (Tedford *et al.* 2019). Early in the reclamation, the lake had a noticeable amount of residual hydrocarbons that produced a sheen, which likely impacted latent energy fluxes by buffering kinetic energy transfer and wave formation or by forming a thin barrier between the water and the atmosphere (Clark *et al.*, 2021). More recently however, lake water quality has improved through management activities such
as mechanical skimming of the surface and the above-mentioned alum treatments.

An eddy covariance platform is ~1 km from the nearest shore so is unlikely to suffer edge effects (Kenny *et al.* 2017). The large moored 6 m by 6 m steel observation platform is stable, except in extreme waves. A full description of the platform and the methods used to derive the eddy covariance observations used to validate the surface models can be found in Clark *et al.* (2021). To ensure high quality data, observations within a 140° arc containing the platform are excluded from this analysis.
Lake temperatures were directly observed at depths of 0.1, 0.2, 0.5, 1, 3, 4, 5, 6, 7, and 8 m at various intervals throughout the study period. To validate the CSLM simulated temperature profile, the observed temperature profile was linearly interpolated in 0.5 m increments from 0.5 to 8 m to align with the lake layers in the CSLM. The mean temperature difference of simulated layer from this interpolated observation record is reported for each depth and as a total water column mean error.

## 2.2 Meteorological forcing

Both models require atmospheric forcing data. Direct observations were used when possible, yet there were missing periods when equipment was malfunctioning or access to the lake platform was limited. Gaps in the meteorological data sets are filled by linear relationships to the NCEP reanalysis 2 product produced by NOAA (Kanamitsu et al. 2002) after first downscaling the reanalysis data to a 30-minute time-step. These datasets are available here: https://psl.noaa.gov/data/gridded/index.html. By default, the CSLM integrates over a 600 second time-step. Therefore, gap-filled observations were linearly interpolated
from 30 to 10 min records for the CSLM simulations. The specific observations used to force the model and the data source are listed in Table 1.

## 2.3 Model descriptions and run settings

The CSLM (MacKay, 2012; MacKay *et al.*, 2017) was refactored to provide fluxes to interpolate missing data from surface-atmosphere eddy covariance based reservoir observations. The CSLM was initially developed to fit into the Canadian Land
Surface Scheme (CLASS; MacKay 2012), to enable the inclusion of lake surfaces in the Canadian Global Coupled Models for climate simulation. Lakes have historically been absent from GCM simulations, and only recently has their inclusion in land surface models been considered (Fisher and Koven, 2020). Briefly CLASS divides each grid cell in the GCM into fractions of surface types and structures to feedback mechanistic behaviours of the earth surface into the atmospheric models. Since CLASS functions on fractional surface area of each grid cell, it is not constrained by the model's resolution. The CSLM is a
1D simulation of the lake surface-atmosphere interaction of any depth divided into user-defined layers. By collapsing the simulation to the vertical dimension, the physics that govern surface temperature through windshear induced mixing, variable light attenuation, buoyancy, and ice development is simplified. When included in larger simulations, the modelled surface



energy exchange is then scaled based on the proportion of open water in the given grid cell back into the atmosphere like in CLASS. This allows the CSLM to work independently or within any simulation that works with CLASS, and produces a more

accurate estimate of ice extent, total cover, and surface temperature of small and mid-sized lakes when compared to the simple water scheme used in Canadian climate models or numerical prediction systems (Garnaud *et al.* 2022).

The FORTRAN 77 codebase for the CSLM has been provided by developer (Murray MacKay, Environment and Climate Change Canada) and contains all the machinery to embed CSLM within CLASS. FORTRAN has limited use outside the earth system model community and older iterations are increasingly difficult to access and compile. Conversely, high-level

languages such as MATLAB and Python are widely taught and applied in earth system sciences and translating the CSLM codebase will allow its application outside climate modelling and numerical prediction systems. Furthermore, in refactoring CSLM, modern programming practices can be implemented, and GOTO and historical design patterns can be replaced with modern structured practices. Although high-level languages often compromise on efficiency with greater overhead in data management, the computational requirements to run CSLM on one lake is low on modern processes (<1 minute per simulated

year on a desktop PC). In addition, refactoring allows unlimited variable length names and whitespaces that make the codebase more intuitive for those who wish to modify or add features (i.e. the name for the latent heat flux from the lake surface is now LatentHeatFlux in both MATLAB and Python compared with HEVLGAT in FORTRAN). A complete list of translated variable names from FORTRAN is listed in Supplementary Table 1.

The CSLM requires the lake latitude, longitude, depth, length, initial temperature profile, instrument observation height, and

extinction coefficient. These parameters can be set via the MakeLakeINI() function in the provided code. For this project, the light extinction varied seasonally and was set to 0.2, 0.3,0.25, 0.2, 0.15, 0.05 and 0.05 for May through November respectively and 0.1 for all other months. These values were determined by running the CSLM with a fixed extinction coefficient at 0.05 intervals from 0.05 through 0.4 and determining the minimum mean squared error of the top 1.5 m of the lake profile from the observed for all years. The code is also set up such that if no variable extinction coefficients are given, the user may specify a

fixed value in the MakeLakeINI() function. At each timestep the option to override the simulated lake temperature profile with direct observations was also added to the CSLM model. If the input observations are missing, not sampled at high enough frequency, or if none are provided, the model will use the simulated profile instead.

The Air-Sea Toolbox (Fairall *et al.* 1996) MATLAB implementation was obtained from the GitHub repository (https://github.com/sea-mat/air-sea). It was run without the cool-skin module and with four of the same atmospheric forcing

variables as the CSLM (see table 1). Surface temperature used in the forcing of the Air-Sea Toolbox was derived from the upwelling longwave radiation and Stefan-Boltzmann Law.

Finally, a five-node artificial neural network (ANN) to estimate evaporation from BML was trained using both models, the same atmospheric variables as required to run the CSLM, the year of observation, and two seasonal vectors. The seasonal vectors are simply $\cos(X)$ and $\sin(X)$, where X is Day-of-year divided by 365 multiplied by $\pi$. The ANN scales the training

data from min/max to -1/1 and a hyperbolic tangent sigmoid is used for the transfer function between the nodes. Training is done through the Levenberg-Marquardt error back propagation. The ANN is trained on 50% of the data, with 25% set aside



for validation and 25% set aside for testing. The adjusted $R^2$ of the ANN output to the testing data was 0.67 for latent heat and 0.72 for sensible.

## 2.4 Analysis

To test the performance of the model in simulating lake dynamics, the model was run continuously from 1 January 2013 to 31 December 2021 without overriding the lake temperature profile. The resulting profile was compared to observations to assess the accuracy of the simulated lake temperature profile.

In the analysis of daily scale observations, days are the sum of the half-hour observations and simulations. Sensible heat flux is in J and the latent heat flux has been converted into mm of water using the average air pressure and temperature of the half-

hour flux. Days with less than 36 half-hour observations were not included in the analysis. Days with less than 48 half-hour observations, but more than 36, had the missing data filled with linear interpolation before calculating the daily flux. Seasonal total evaporation was calculated for the CSLM, and Air-Sea Toolbox, simulated latent heat flux as well as the ANN gap filled observed latent heat flux.

Comparisons of half-hourly, and daily modelled sensible and latent heat fluxes are done with ordinary least squares regression

(OLS). Seasonal bias was evaluated by plotting the coefficient of determination (R2) from OLS preformed on data grouped by each month of the year. In addition to the R2, for more direct visual inspection, the mean daily evaporation rate for all years was plotted by DOY for all simulated and observed fluxes. So as not to change the sample depth for the mean calculation between modelled and observed, only days when an observed flux was available were used for all three plots.

Throughout this analysis the term 'error' was used explicitly as the observed value minus the modelled value divided by the

observed value. This prevented the error analysis from being skewed towards the largest fluxes. A boxplot was generated for the error from each model in each month of the year. Errors are then evaluated with respect to the state of the atmosphere (Total Kinetic Energy, TKE; friction velocity, $u_*$; Obukhov length, OL; and vapour pressure deficit, VPD). For visual clarity, error terms are binned to the corresponding atmospheric state variable and plotted as boxplots. The bins are determined by 5 percentile ranges, such that the first bin is all the fluxes that correspond to < 5th percentile, the second bin is ≥5th through <10th

and so on. Analysis on the error was also conducted using windspeed, as windspeed is a widely measured variable unlike the more micro-meteorological specific variables described above. For simplicity in the windspeed error analysis the error was grouped in 1 m s$^{-1}$ bins instead of percentiles.

Finally, to evaluate the ice-module of the CSLM, ice free days were compared to the observed days of ice-on and ice-off when it was available. Then the total evaporation was compared to the number of predicted ice-free days along with, average daily

temperature, average relative humidity, and average windspeed for the same period. OLS was used to assess any relationship between evaporation rates and ice-free periods, and average temperature.



## 3 Results

When run without the lake temperature observations overriding the simulation, the CSLM modelled temperature of the water column overestimates the observed water column by $1.3 \pm 1.3°C$ for all the periods and depths water column temperatures are available (Fig. 1; Table 2). However, the observations are not evenly spread across the season and August and September are more represented than May, June, and July periods. The upper 3 m of the simulated profile is much closer to the observations (mean temperature difference of $0.6 \pm 1.0°C$) than the deeper layers (mean temperature difference of $1.6 \pm 1.3°C$ for 3.5 to 8 m) (Fig. 1; Table 2). Fall turnover is well captured by CSLM with respect to the lake temperature profile, but the simulation is still warmer (mean difference of $1.2 \pm 0.6°C$ for 01 September to 31 October).

The half-hour latent heat fluxes from the CSLM and the Air-Sea Toolbox approximated the observed fluxes ($R^2$ 0.60 and 0.56 respectively, p-value < 0.001; Fig. 2). However, the ANN increased the accuracy substantially ($R^2$ 0.69, p-value < 0.001). The modelled daily total evaporative flux was closer to the total observed daily flux for all the models ($R^2$ 0.76, 0.83 and 0.91 for Air-Sea Toolbox, CSLM, and ANN respectively; Fig. 2). The models are better at approximating the larger scale trends and have better accuracy at the daily scale (as measured by $R^2$) and slopes closer to 1 (Fig. 2). Like the lake temperature profiles, the half-hour (and daily) estimates generally improved in recent years compared to the early years (Table 3). In relative units, the root mean squared error for the daily estimated evaporation rate for the whole study period was 0.83, 0.71, and 0.52 mm day$^{-1}$ for the Air-Sea Toolbox, CSLM and ANN respectively (see Table 3 for interannual breakdown).

Estimated sensible heat flux followed a similar trend to the latent heat flux at the half-hour scale ($R^2$ 0.57, 0.55, and 0.71, p-value < 0.001 for Air-Sea Toolbox, CSLM, and ANN estimates respectively). At the daily scale, the modelled fluxes more accurately estimated the observed flux variability ($R^2$ 0.83, 0.6, and 0.94, p-value < 0.001 for Air-Sea Toolbox, CSLM, and ANN estimates respectively) but only the ANN estimated sensible heat fluxes were close to 1:1 (Fig. 2). Notably, in 2019 and 2020 there were several near-zero estimated sensible heat fluxes when the eddy covariance observations spanned the whole range of reasonable values (Fig. 2). The comparison of the accuracy between modelled and eddy covariance observed data can become problematic in certain atmospheric conditions and is discussed more below.

Each model's performance also has a seasonal bias. Frozen season fluxes are poorly estimated, but that may be a consequence of limited observations to compare too (Fig. 3 and Fig. 4). Estimated latent heat fluxes in the spring (March, April, and May) have the lowest relative accuracy of the year for all models (Fig. 4). Early warming period (May for BML) also has the lowest $R^2$ values, which increase throughout the ice-free season (Fig. 3). The relative error in the spring is somewhat reduced, and the median is closer to zero for the ANN estimated fluxes, but the error is still much higher in the spring than the rest of the year. These low relative errors do not accumulate substantially in the total seasonal evaporation rates as spring fluxes are quite low compared to those in the fall (Fig. 4). The coefficient of determination between estimated and observed fluxes improves throughout the year until ice onset sometime in November (Fig. 3). The CSLM tends to predict higher surface-to-atmosphere latent heat transfer during the warming phase, and lower during late fall before ice onset (Fig. 4).



The typical accuracy in response to the state of the atmosphere was consistent. Both the CSLM and the Air-Sea Toolbox
215   estimated fluxes had much lower accuracy when there was limited kinetic energy in the atmosphere (i.e. when the atmosphere
was stable). Low total kinetic energy, friction velocity, vertical and cross wind variance (not shown) as well as positive near
zero Obukhov Length increased the error of the model output compared to the observed fluxes (Fig. 5). The ANN did not
provide much improvement during these conditions and showed median errors in excess of the other models when there was
limited mixing (low TKE, $u_*$). There was a slight increase in the median error as a function of $u_*$ (Fig. 5), suggesting that at
220   high friction velocities the models underestimated the latent heat flux. All three models were relatively accurate at high VPD
when evaporation is highest but had greater error in humid conditions (Fig. 5). For more general interpretability outside of a
micrometeorological analysis, the error associated with atmospheric turbulence and stability can be approximated by low
windspeeds (Fig. 6). At high windspeeds the models tend to overestimate the rate of evaporation and at low windspeeds they
underestimate it (Fig. 6). It should be highlighted that the median ANN latent heat flux compensates for this error because
225   windspeed is part of the training data for the ANN and therefore has already adjusted the latent heat flux based on the typical
offset unlike the above micrometeorological variables. However, the variance in the error from the ANN simulated heat flux
is still high at low windspeeds (Fig. 6).

Annually the CSLM and Air-Sea Toolbox was within 11 ± 85 mm and 27 ± 54 mm of the seasonal evaporative flux of 452 ±
58 mm (Fig. 7). In addition to the high seasonal accuracy, when run continuously for the whole 2013-2021 periods, using the
lake profile data when available, the CSLM simulates the length of the ice-off season within a few days. Specifically, the
CSLM predicted ice-off 0.4 ± 5.9 days early and 2.9 ± 2.2 days early for ice-on, for an average extra 3.0 ± 5.9 days of open
water (Table 4).

## 4 Discussion

The first objective of refactoring CSLM now has code available on GitHub as open-source codebases in both python
(https://github.com/mclark04/CSML_Python) and MATLAB (https://github.com/mclark04/CSLM_MATLAB). They have
been written such that the logic is more consistent with modern high level coding patterns and the variable names are all in
camel case to make it more intuitive. The second objective of assessing the modelling approaches is more complex and below
we discuss the advantages and disadvantages of the three modelling approaches.

BML is an oil sands pit-lake that is highly managed, has had variability in turbidity that spans orders of magnitude, has been
impacted by hydrocarbons, and has tailings, not mud or bedrock as its bottom. Although the turbidity has become lower and
less variable since the alum treatment in 2016, the large variability in turbidity during the early years likely caused the larger
errors in the top 3 m of the temperature profile and the resulting latent energy fluxes simulated by the CSLM. The refactored
code can accommodate a continuous extinction coefficient, but the turbidity data for BML was not continuously available. In
the early years, BML had a notable hydrocarbon sheen (Clark *et al.* 2021). Hydrocarbons on water can create a barrier between
the atmosphere and the water surface and momentum (sheer) transfer into the water (i.e. waves). Exactly how this impacts the



latent energy flux for a mid-sized lake is an area of ongoing research, but it likely contributed to the early years error in the surface atmosphere exchange. Finally, there are two irregular conditions in BML that may cause bias that other lakes do not have. First, below the mud layer in BML there are fluid fine tailings, not sediment/bedrock as assumed in the CSLM. The impact of having a semi-fluid here could produce an energy leak to the underlying tailing that is not reflected in the CSLM.

Second, there is some uncertainty on the heat budget due to active inputs of warm tailings (~60°C) in the northeast corner of BML. Both of these conditions are likely to have negligible impacts on the surface heat exchange but are worth noting when comparing future work.

Despite all the factors unique to BML, variation in the extinction coefficient remains the largest source of uncertainty within the CSLM with respect to estimating the BML atmospheric heat exchange. It is very likely that the increased stability in the

turbidity at BML led to more accurate simulated temperature profiles in the later years (Fig. 1). Generally, all three models performed better after the alum treatment in 2016. Variable turbidity can be accounted for in the CSLM, if observations are made periodically, such as at the time of site maintenance visits. However, turbidity data was not available at the time of analysis for this study. The second largest discrepancy of modelled fluxes compared to observed fluxes occurred during periods of high atmospheric stability (Fig. 5). In general, both models underestimated latent heat flux at windspeeds $< 2$ m s$^{-1}$ (Fig. 6).

Czikowsky *et al*. (2018) suggested that a lower limit of 4 m s$^{-1}$ is a better threshold to separate meso-scale from localized circulation, but here both models perform well at half that windspeed. At very low windspeeds however, due to the assumptions in the eddy covariance method, it is difficult to separate errors in modelling over errors in observation. The ANN model had greater accuracy with respect to the observed fluxes at low windspeeds and/or stable conditions, but again due to the limits on EC theory it is difficult to say if this is more reflective of the lake-atmosphere exchange or simply the neural network biasing

towards potentially inaccurate observations. Furthermore, the 1-D simulation of lake physics is likely flawed in calm conditions. When the shear force is low, buoyancy forces dominate near-surface circulation, and the 1-D simulation inaccurately captures the smaller convective processes in the water column. Under stable conditions, smaller convective processes can produce significant spatial variability in surface temperatures (Tedford *et al*. 2014).

Overall, the CSLM performs well for BML. It accurately reflects both fall turnover (Fig. 1) and ice melt and onset (Table 4).

Both models underperform at predicting the half-hour heat flux observations but are much more effective at simulating the daily (Fig. 2) and annual evaporation rates (Fig. 7). The CSLM out preforms the Air-Sea Toolbox at all scales (Fig. 4), but heat flux estimates are improved with a neural network that in essence fine tunes the model output to the observed fluxes between the lake and the atmosphere. Generally, the estimated heat fluxes improved over time, and were more accurate in the cooling phase than in the warming phase (Fig. 3). Fournier *et al*. (2021) found that bulk transfer estimated evaporation, calculated with local meteorological observations, were between 1.38 and 0.62 mm day$^{-1}$ for one year of observations in each

of two different boreal reservoirs (located in Quebec, Canada). The lower end of this range overlaps with the root mean squared error from the Air-Sea toolbox at BML (RMSE 0.83 mm day$^{-1}$, see Table 3). The CSLM had a slight improvement (RMSE 0.71 mm day$^{-1}$) but the ANN based estimate (RMSE 0.52 mm day$^{-1}$) was an improvement over all the evaporation estimates presented by Fournier *et al*. (2021). Both CSLM and the Air-Sea toolbox did a poor job of estimating the observed heat



exchange in the frozen months, but particularly December when observations were very sparse (Fig. 3). It may be better to use
more traditional gap filling methods when the ice cap is present and observations are limited, as the surface is more similar to
a rigid terrestrial surface. Furthermore, it is likely that the improvement of the models to estimate both energy fluxes over the
study years is due to the decreased variability in turbidity and decreased total turbidity of BML in recent years. This is
promising for applications of the CSLM to estimate evaporation losses across a large range of managed water bodies and
reservoirs. In general, the CSLM is good for evaporation estimates at daily and monthly scales typically used in water
management and the ANN provides additional accuracy for gap filling micrometeorological research networks.

## 5 Conclusion

We have made the CSLM open source in two widely used languages in addition to refactoring the FORTRAN code to be more
accessible. The refracted CSLM, in both MATLAB and Python, are available with documentation on GitHub (here) with the
aim of providing an open-source repository for other managers and scientists interested in lake model simulations. One year
of BML observations and atmospheric data are also provided in the GitHub repository so users can be sure they are running
the simulation correctly.

The CSLM works well for gap filling missing data from eddy covariance lake observation systems and is more accurate relative
to the eddy covariance system than the Air-Sea Toolbox bulk flux estimates used in this study. The CSLM performs poorly in
stable and/or humid conditions at the half-hour scale, but the evaporation estimates accurately reflected the daily scale EC
observations. The flux estimates can be further improved with a simple neural network requiring only basic understanding of
machine learning techniques. Furthermore, running the CSLM in addition to direct observations has the advantage of providing
information about important lake processes that may not have been directly observed (such as ice onset/melt, thermocline
depth, and turnover timing).

We believe this will be useful for managers who need to estimate the evaporative fluxes from a northern lake system. We
found that the biggest hurdle in using the CSLM will be obtaining a reasonable estimate of the light extinction coefficient, but
this is likely more stable in most managed water bodies than in the pit-lake under study here. Most researchers and mangers
should be able to estimate the light extinction coefficient with very minimal time or cost commitments during any site visit.

## 6 Code and data availability

As mentioned above, the refactored CSLM code and one year of data can be accessed here:
https://github.com/mclark04/CSLM_MATLAB (MATLAB) and https://github.com/mclark04/CSML_Python (Python).
The MATLAB scripts for analysis and plotting as well as the full dataset to reproduce this paper are available at the following
DOI: 10.5281/zenodo.10470869




## 7 Author contributions

SKC supervised MGC and acquired funding. MGC refactored the FORTAN code and developed the git repositories. MGC ran the CSLM and neural networks and performed the analysis. MGC prepared the manuscript with contributions from SKC.

## 8 Competing interests

This work was partially funded by an industrial partner in the oil and gas sector of Canada.

## 9 Acknowledgements

We would like to thank Mike Treberg and Gordon Drewitt for all their site visits and hard work to maintain our lake platform. We would also like to thank Sycnrude Canada Ltd, and specifically the Closure and Reclamation Department who have been supportive of this research. We also acknowledge support of the Global Water Futures program.

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



## 11 Tables and Figures

**Table 1: Meteorological variables used in forcing the two models. Precipitation has no R2 as the reanalysis value was used directly when there was gaps in the local data.**

| Canadian Small Lakes Model | Air-Sea Toolbox | Reanalysis product used to gap fill. | $R^2$ to original trace |
|---|---|---|---|
| Windspeed | Windspeed | uwind.10m.gauss.YYYY.nc vwind.10m.gauss.YYYY.nc | 0.23 |
| Air Temperature | Air Temperature | air.YYYY.nc | 0.93 |
| Specific humidity | Relative Humidity | shum.2m.gauss.YYYY.nc | 0.1 |
| Air Pressure | Air Pressure | pres.nlog.sfc.spec.YYYY.nc | 0.91 |
| Downwelling Shortwave | N/A | dswrf.sfc.gauss.YYYY.nc | 0.23 |
| Downwelling Longwave | N/A | dlwrf.sfc.gauss.YYYY.nc | 0.75 |
| Precipitation | N/A | prate.sfc.gauss.YYYY.nc | N/A |






**Table 2: Mean temperature difference between observations and each lake layer of the Canadian Small Lakes Model (CSLM) run without overriding lake temperatures. Negative values indicate that the CSLM over estimated temperatures while positive values indicate an underestimate.**

| Depth (m) | May | June | July | Aug | Sept | Nov | Annual |
|---|---|---|---|---|---|---|---|
| 0.5 | -0.2 ± 2.5 | 0.6 ± 1.6 | 0.3 ± 1.2 | -0.6 ± 0.7 | -0.9 ± 0.6 | -0.9 ± 0.3 | -0.4 ± 1.2 |
| 1 | -0.9 ± 1.5 | 0.1 ± 0.9 | -0.0 ± 0.9 | -0.8 ± 1.8 | -1.2 ± 0.9 | -1.2 ± 0.4 | -0.8 ± 1.3 |
| 1.5 | -0.7 ± 1.4 | 0.1 ± 0.8 | -0.1 ± 0.7 | -0.8 ± 0.6 | -1.2 ± 0.8 | -1.1 ± 0.4 | -0.7 ± 0.9 |
| 2 | -0.6 ± 1.3 | -0.0 ± 0.8 | -0.2 ± 0.7 | -0.8 ± 0.5 | -1.1 ± 0.8 | -1.1 ± 0.4 | -0.7 ± 0.8 |
| 2.5 | -0.6 ± 1.4 | -0.1 ± 0.9 | -0.3 ± 0.7 | -0.8 ± 0.5 | -1.0 ± 0.7 | -1.0 ± 0.3 | -0.7 ± 0.8 |
| 3 | -0.6 ± 1.5 | -0.1 ± 1.2 | -0.4 ± 0.8 | -0.8 ± 0.5 | -1.0 ± 0.6 | -1.0 ± 0.3 | -0.7 ± 0.8 |
| 3.5 | -0.7 ± 0.8 | -0.6 ± 0.9 | -0.8 ± 0.9 | -1.0 ± 0.5 | -1.1 ± 0.5 | -1.0 ± 0.5 | -0.9 ± 0.7 |
| 4 | -1.7 ± 0.7 | -1.3 ± 1.1 | -1.3 ± 1.1 | -1.1 ± 0.7 | -1.1 ± 0.6 | -1.1 ± 0.6 | -1.2 ± 0.8 |
| 4.5 | -0.5 ± 0.9 | -1.4 ± 1.2 | -1.9 ± 1.3 | -1.5 ± 1.0 | -1.5 ± 0.8 | -1.3 ± 0.5 | -1.5 ± 1.0 |
| 5 | 0.7 ± 1.5 | -1.6 ± 1.6 | -2.3 ± 1.8 | -1.8 ± 1.7 | -1.9 ± 1.3 | -1.6 ± 0.7 | -1.7 ± 1.6 |
| 5.5 | 0.2 ± 1.1 | -2.0 ± 1.1 | -2.9 ± 1.4 | -2.4 ± 1.5 | -1.9 ± 1.2 | -1.5 ± 0.6 | -2.1 ± 1.4 |
| 6 | -0.5 ± 1.0 | -2.3 ± 1.0 | -3.4 ± 1.2 | -2.9 ± 1.5 | -1.9 ± 1.1 | -1.5 ± 0.6 | -2.4 ± 1.4 |
| 6.5 | -0.3 ± 1.1 | -1.6 ± 1.5 | -3.4 ± 1.9 | -3.2 ± 1.6 | -2.1 ± 1.4 | -1.6 ± 0.8 | -2.4 ± 1.7 |
| 7 | -0.1 ± 1.4 | -1.0 ± 2.3 | -3.2 ± 2.8 | -3.5 ± 2.1 | -2.2 ± 1.7 | -1.6 ± 0.9 | -2.4 ± 2.3 |
| 7.5 | -1.1 ± 1.0 | -0.8 ± 1.5 | -3.0 ± 2.0 | -3.3 ± 1.6 | -1.7 ± 0.7 | -1.4 ± 0.4 | -2.2 ± 1.6 |
| 8 | -2.0 ± 0.7 | -0.4 ± 1.5 | -2.4 ± 1.8 | -2.7 ± 1.8 | -1.0 ± 1.1 | -1.2 ± 0.7 | -1.7 ± 1.7 |
| All | -0.6 ± 1.5 | -0.7 ± 1.5 | -1.6 ± 2.0 | -1.7 ± 1.6 | -1.4 ± 1.1 | -1.2 ± 0.6 | -0.9 ± 0.7 |






**Table 3: Root mean squared error (mm day$^{-1}$) between observations and Air-Sea toolbox (StA), Canadian Small Lakes Model (CSLM), and artificial neural network (ANN) respectively. Number of observations varied each year.**

|  | StA | CSLM | ANN |
|---|---|---|---|
| 2013 | 0.71 | 0.64 | 0.60 |
| 2014 | 0.74 | 0.68 | 0.77 |
| 2015 | 0.92 | 0.80 | 0.58 |
| 2016 | 0.84 | 0.70 | 0.63 |
| 2017 | 0.69 | 0.55 | 0.41 |
| 2018 | 0.75 | 0.65 | 0.44 |
| 2019 | 0.45 | 0.55 | 0.36 |
| 2020 | 0.52 | 0.48 | 0.45 |
| 2021 | 0.77 | 0.57 | 0.17 |
| 2022 | 0.81 | 0.90 | 0.31 |
| Mean (±SD) | 0.72 (±0.14) | 0.65 (±0.13) | 0.47 (±0.18) |





**Table 4: The Canadian Small Lakes Model (CSLM) simulated vs observed day of ice breakup and ice onset. Average accuracy of the CSLM is 0.1 ± 6.1 days early for ice off and 2.4 ± 2.1 days early for ice on. Ice free days is calculated from the CSLM data.**

| | CSLM | | Observed (Syncrude) | | Average difference (obs-modled) | | Ice free days |
|---|---|---|---|---|---|---|---|
| | Ice OFF | Ice ON | Ice OFF | Ice ON | Ice OFF | Ice ON | |
| 2013 | 05-May | 09-Nov | N/A | 10-Nov | N/A | 1 | 188 |
| 2014 | 11-May | 12-Nov | 01-May | 11-Nov | -10 | -1 | 185 |
| 2015 | 22-Apr | 19-Nov | "late April" | 20-Nov | N/A | 1 | 211 |
| 2016 | 30-Apr | 18-Nov | 27-Apr | 18-Nov | -3 | 0 | 202 |
| 2017 | 01-May | 06-Nov | 05-May | 08-Nov | 4 | 2 | 189 |
| 2018 | 28-Apr | 06-Nov | 05-May | 08-Nov | 7 | 2 | 192 |
| 2019 | 21-Apr | 06-Nov | 20-Apr | 11-Nov | -1 | 6 | 198 |
| 2020 | 06-May | 05-Nov | 06-May | 10-Nov | 0 | 9 | 179 |
| 2021 | 30-Apr | 15-Nov | 06-May | 21-Nov | 6 | 6 | 199 |







**Figure 1: Difference between observed and Canadian Small Lakes Model (CSLM) estimated BML lake temperature profile for when the CSLM was run without overriding the lake temperatures with the observed temperatures.**






**Figure 2: Scatter plots of the modelled and observed heat fluxes at both the half-hour and daily scales for Air-Sea toolbox (StA), Canadian Small Lakes Model (CSLM), and artificial neural network (ANN) respectively. Dashed lines are 1:1 lines, solid lines are regression lines. Top row is half-hour latent energy fluxes (LE), second row is daily latent energy fluxes, third row is half-hour sensible heat fluxes (H) and bottom row daily sensible heat flux. $R^2$ and the slope of the regression line (β) is given above each plot. All regressions are significant with p-values < 0.001.**



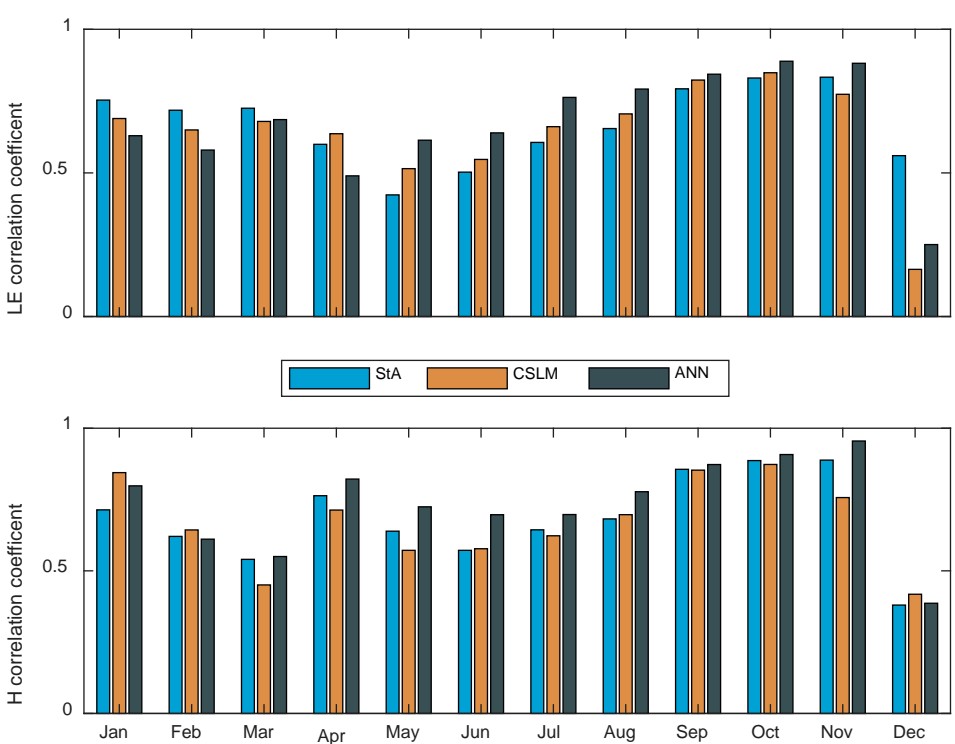

**Figure 3: Coefficient of determination between model predicted latent/sensible heat flux (top/bottom) for each month pooled across all years (left) for Air-Sea toolbox (StA), Canadian Small Lakes Model (CSLM), and artificial neural network (ANN) respectively. All values are significant (p-values <0.0001) except the latent heat fluxes from CSLM in December (p-value = 0.52), ANN in Jan (p-value = 0.01).**






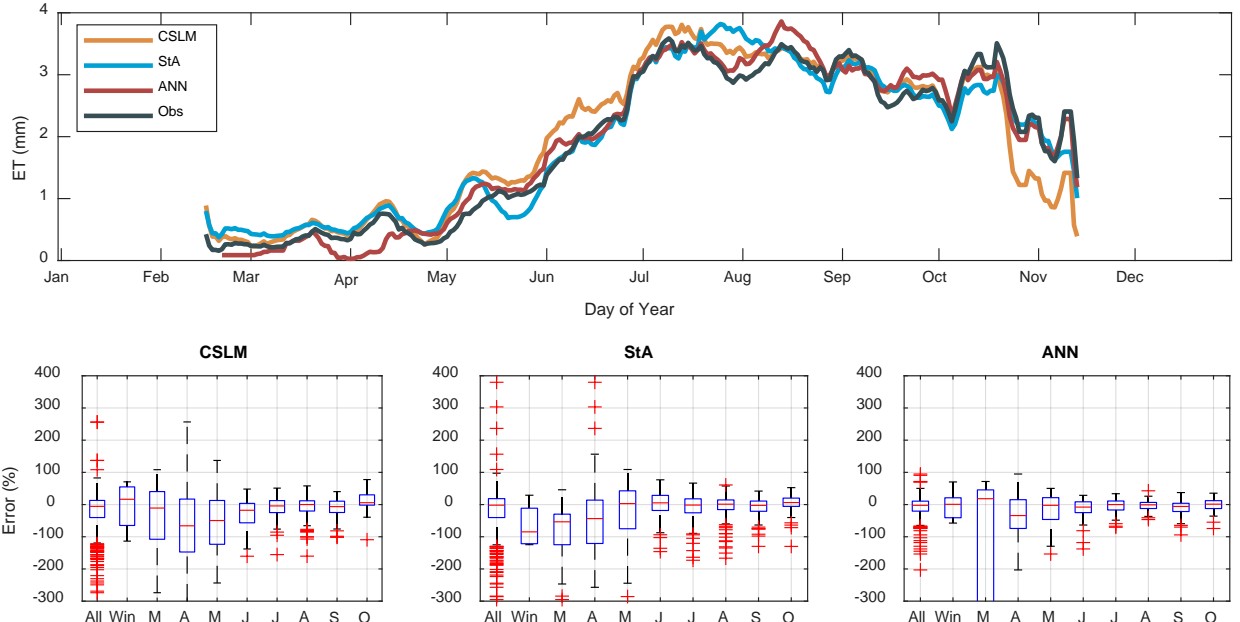

**Figure 4: Top, the average daily flux over the nine years of observations. Bottom monthly error rates of the models (Observation – Estimation). In the top figure, for comparability the Air-Sea toolbox (StA), Canadian Small Lakes Model (CSLM), and artificial neural network (ANN) estimated evaporation rates were only included if there way a corresponding daily observed value.**






**Figure 5: Error [(Observed-Modeled)/Observed x 100%] as a function of atmospheric turbulence, stability, and vapour pressure deficit. Solid line is median value, dashed lines cover the interquartile range.**



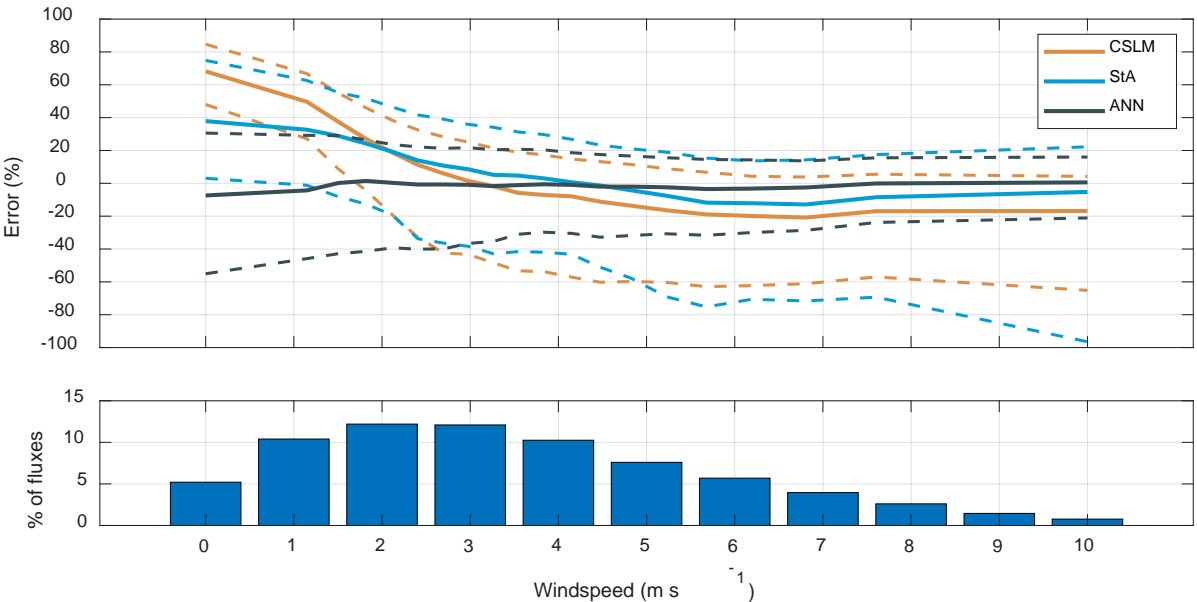

**Figure 6: Top is model error and bottom is a histogram of the fluxes as grouped by windspeed (m s$^{-1}$). Dashed lines represent the interquartile range and solid line is the median error.**





**Figure 7: Cumulative sum (mm) of evaporation from BML. For the cumulative sum of the observed fluxes, the gaps were filled with the ANN model.**