# Peer review of "An open source refactoring of the Canadian small lakes model for estimates of evaporation from medium sized reservoirs"

_EGUsphere, 2023_

## Author Comment (AC2)

[Figure]

Figure S1: Standardized feature performance (mean squared error) from permutation analysis on Artificial Neural Networks used for gap filling latent heat (left) and sensible heat (right).  Higher value implies greater error associated with randomizing the sequence of the input feature.

---

## Author Comment (AC3)

[Figure]

Supplementary Figure 2: Base Mine Lake, platform location, and surrounding area. Google Earth.

---

## Author Response (AR1)

**Reviewer 1**

This study proposes the use of the Canadian Small Lake Model as a means of estimating evaporation from medium-sized reservoirs. The paper is organized around two main objectives. The first is to refactor the Canadian Small Lake Model code, originally written in FORTRAN77, into high-level programming languages, namely MATLAB and Python. The second is to evaluate the evaporation estimates produced by the CSLM by comparing them with direct observations on a pit lake in northern Alberta, and to put them in perspective with other commonly used approaches (e.g., bulk transfer).

Evaporation remains a difficult hydrological flux to quantify, especially on water bodies. The authors' approach to facilitate access to a physically based model such as the CSLM for this purpose is of great relevance. The applications of the CSLM obviously go beyond the evaporation process alone. I have few comments which, once considered, should make the article more convincing and facilitate its acceptance.

**Main comments**

1) Refactoring

What steps were taken to ensure that CSLM in Python, MATLAB and FORTRAN77 behave absolutely identically?

We compared the surface film temperature.  They do not behave absolutely identically, there is slight variability among them all yet the differences are extremely small.  It is highly unlikely the numerical solutions would ever yield exact results.  The difference occurs in the precision of the 32-bit floating point estimates of buoyancy and sheer forces. Python and MATLAB carry more overhead in bits than FORTRAN, so less data for the integer values in the numerator and denominator in the fraction approximating the decimal. However, Dr. Murray MacKay who developed the FORTRAN code provided output from his experimental data that was used on the Environment and Climate Change Canada cluster with the original code and we compared that with the FORTRAN run on a local machine. Our error is within the variance observed between running the FORTRAN code on my machine and the difference observed between other languages.  That is how we determined the code was ported successfully.

2) 1D hypothesis

What is the approximate importance of advective fluxes in the energy balance of the Base mine lake if it is "heavily managed"? Does the 1D hypothesis of the CSLM model hold up? I understand that the authors cannot investigate this aspect in detail, but its importance should at least be mentioned and used as a discussion element for some of the plots.

This is a good question and as stated by the reviewer it would be very difficult, if not impossible, to determine specifically.  In the northwest corner the lake water is extracted for mine use, and there is input of process water from industrial use into the tailings layer below the lake. We have not observed any major deviation in the lake model temperatures compared to the observed, so it is likely this heat is minor with respect to the volume of the lake itself. We added the sentence: "The volume of water

additions (sourced from a nearby lake with a similar surface temperature) are unlikely to significantly impact the overall energy budget due to the relative volume of input water (mean of 3.2% of the total lake volume and declining over the study period)" to the study site description. In the original manuscript, we have a section in the discussion speculating about the implications of warm tailings additions under the lake.

3) Lack of energy balance closure

Eddy covariance measurements are known to underestimate the surface energy balance. For terrestrial environments, it would be unthinkable to set up an EC analysis without accounting for this. I do not think a consensus has emerged in the aquatic community on how to manage this issue. Is it possible to mention it minimally and think about how to proceed?

Agreed that there is no consensus on this issue. Even on a non-ideal terrestrial system it is challenging, let alone on a fluid surface that can exchange momentum as well as heat with the atmosphere. We decided to expanded the discussion around the error analysis to explore this. Since the model and the observations are most dissimilar under atmospheric conditions that are unfavorable for eddy covariance systems, that is where we added this discussion the reviewer requests. Specifically, we added the following two sentences to the discussion: "The "energy closure" issue (potential underestimate in the total of sensible and latent heat fluxes) in terrestrial environments with EC systems has been extensively debated in the literature, but the authors are unaware of discussion about how to evaluate energy closure accuracy in lake systems due to the complexities introduced by a fluid water column. Here the CSLM may help future analysis by providing models of the underlying processes (primarily the sheer and buoyancy fluxes) within the water column during times of surface heat discrepancy."

4) ANN

I suggest doing a permutation importance analysis on the ANN model

Thank you for this suggestion. We have included this additional analysis in the supplemental material. It was placed in the supplemental material so as not to distract from the CSLM discussion with nuance around the ANN, but it is there now for reference. The reader is directed to it in the methods: "A permutation importance analysis is included as supplementary Figure 1."

**Minor Comments**

Section 2.1: Is it possible to provide an overview of the limnological behavior of the water body somewhere, e.g., by providing the longevity of the ice cover, dates of freeze-up and break-up, etc.?

Added the text "Over the study period, BML typically had a total ice cover by early-to-mid November and it melted by late April/early May" into the site description in section 2.1 immediately following the climate description. The exact dates of observations are provided in Table 4.

l. 91-92: The platform operated all year round, right? Say so explicitly. Also, describe the platform and how the fluxes were determined; it is not enough to just refer to Clark et al. (2021).

We have expanded the description of the flux processing in this section to include typical micrometeorological literature and processing methods of the EC fluxes.

Section 2.4: This section is messy. Is it possible to revise the structure, especially defining subsections?

Thank you for the feedback. We broke the section up into 4 subsections. Computation of daily fluxes, comparisons of simulated and observed fluxes, error analysis, and ice simulation analysis.

l. 172: Does TKE really mean total kinetic energy? Since friction velocity and Obukhov length are mentioned right after it, I would be tempted to think the authors mean Turbulent Kinetic Energy. Please double check.

Yes, you are correct. Thank you for noticing that.

l. 186: It is mentioned here that the first 3 m of the simulated profile are much closer to the observations than the bottom temperatures. The authors should remind readers that the calibration of the extinction coefficients was performed on the first 1.5 m of the temperature profile.

Agreed, this should be clear. We added the sentence "It should be noted that the extinction coefficient was calibrated on the 1.5 m depth, so likely has to do with the accuracy over this region."

**Reviewer 2**

**GENERAL COMMENTS**

The authors in this manuscript refactored the Canadian Small Lakes Model (CSLM) into modern high-level programming languages in both python and MATLAB in open-source repositories, and evaporation estimates by the CSLM were assessed using nine years of EC observations of a highly managed pit-lake in Northern Alberta. The reprograming may be very important for improving the Canadian Global Coupled models. However, as to be possible publication in an international journal, the reviewer suggest that the authors discuss the results and its performances in a wider scope for different type of lakes all over the world.

We appreciate the time of reviewer 2 and the reflection in this review. Unfortunately, we believe it is outside the scope of this article to compare the functioning of the CSLM to a multitude of lakes globally. Eddy covariance data is very difficult to obtain over water bodies and such multi-region study would be extremely challenging. Our examination of the AmeriFlux database revealed few sites with requisite flux and lake data to test this model. We also do not advocate for improving the Canadian Global Coupled models (that has already been done in Verseghy and MacKay (2017) and for numerical prediction systems in Garnaud et al. (2022)). As stated, we aim to improve provide a second test of its validity and improve the access of the CSLM for use by water resource managers who need accurate evaporation measurements of the bodies of water under their charge. Globally, dams and reservoirs are increasing at an alarming rate, and tools for practitioners who cannot run earth system models are desperately needed.

In addition, the authors should provide more details about their method and study areas. What is the Canadian small lakes model? Why should it be only used in small to medium sized lakes? Furthermore, the authors have adopted a highly managed pit-lake that is not a naturally lake as stated by the authors. Hence, a more in-depth discussion may be needed, i.e., to what possible extent these artificial effects and conditions could impact on the simulated results?

As suggested by Reviewer 1, we have increased the description of the EC methodology in the revised manuscript. The limitations on large lakes is primarily due to advective transport, but that would also

pose a great challenge for the EC theory and is far beyond the scope of this paper. While we agree that the lake is highly managed, and we have increased the discussion of the management in response to Reviewer 1's comments, we do not think it is fundamentally unique with regards to the surface-atmosphere exchange.

**SPECIFIC COMMENTS**

Line 32: …when the atmosphere is generally warmer than the air. Shall here air be water surface of lake?

Thank you for catching this. Yes, when the atmosphere is generally warmer than the lake. Changed.

Line 67: Is the Air-Sea Toolbox the so-called bulk transfer method?

Yes, changed bulk transfer above (line 55) to bulk **flux** transfer for clarity.

Line 70: why high level languages was mentioned repeatedly in the introduction? Do you hope to provided interface-friendly software for scientists and managers?

No we have no plans for interface-friendly software. However, Python and MATLAB are more relevant and accessible than FORTRAN to researchers now. Most Earth/Hydrology/Environmental Sciences graduate programs now teach high-level languages such as Python, MATLAB, or R to their graduate students, whereas FORTRAN is becoming increasingly limited to earth system modelling communities.

Lines 75-98: a site map is needed to show where the lake as well as observations (i.e., the eddy covariance platform) are located. In addition, the author could be best to provide another diagram that show clearly the complex structure of the pit-lake and different depth of temperatures observations.

Thank you for this feedback. We omitted a site map to limit the number of figures. The platform is in the center of a mostly round lake as described in the site description (>1km from any shore). Since there is not much to see but wanting to address this comment we put a figure into the supplementary material under supplementary figure 2. As for the complex structure of the pit-lake, the lake structure is in fact quite simple. It was built to be a large bowl-shaped feature on the landscape. The bottom of the bowl was a tailings pond and then freshwater was pumped over the tailings surface. References are provided to papers which deal with the design and performance of the design, but we think since it has little impact on the surface-atmosphere exchange that such a discussion is beyond the scope of this paper.

Lines 128-129: Although the computational requirements to run CSLM on one lake is low, however, if it was embedded into the GCM what is about the increasing computational time.

The use of the CSLM in GCM is not suggested anywhere in the manuscript, and has little to nothing to do with the results of this paper. Much has been already written on that subject and we refer the reviewer to papers on this topic by Verseghy and MacKay (2017).

Line 135: what is extinction coefficient? It should be defined upon its first occur in the context.

Thank you for highlighting this. The extinction coefficient is the physical property of translucent substance to absorb light with depth. We added "(how rapidly light is absorbed in the water column)" to its first use for readers unfamiliar with the concept.

Lines 139-140: 'The code is also set up such that if no'. Has it been set up as an interface that facilitates above selections? Plz provide the related interfaces.

Sorry for the ambiguity but there are no interfaces provided with this code. All settings are passed as arguments to the functions listed. All arguments, and their default settings if not specified, are defined at the top of the functions in the code itself. This code is not written for any user interfaces (UI) and expects baseline coding knowledge to operate.

Line 156: Do you mean overriding the lake temperature profile with simulation results?

No, we mean that the simulation derived the lake temperature profile. This was included so as not to confuse the reader after stating that the model COULD be run with actual lake temperature profile observations if they are available.

Line 170: what is the boxplot? Plz provide the related interfaces.

A description of the boxplot was added to the figure "Boxplots show the median (red line), inter quartile range (blue box), 5th to 95th percentile range (whiskers), and outliers (red +)."

Lines 239-252: The study lake is highly managed as the authors have stated. So the reviewer suggested that could you discuss to what possible extent these two irregular conditions could impact on the surface heat exchange and the simulated results?

Thanks for mentioning this. We have hopefully addressed this in response to Reviewer 1.